# Interaction of Substrates with γ-Secretase at the Level of Individual Transmembrane Helices—A Methodological Approach

**DOI:** 10.3390/ijms241814396

**Published:** 2023-09-21

**Authors:** Theresa M. Pauli, Ayse Julius, Francesco Costa, Sabine Eschrig, Judith Moosmüller, Lea Fischer, Christoph Schanzenbach, Fabian C. Schmidt, Martin Ortner, Dieter Langosch

**Affiliations:** Lehrstuhl für Chemie der Biopolymere, Technische Universität München, Weihenstephaner Berg 3, 85354 Freising, Germany; theresa.pauli@tum.de (T.M.P.); ayse.julius87@gmail.com (A.J.); fcosta@ebi.ac.uk (F.C.); sabine.eschrig@tum.de (S.E.); moosmueller.judith@gmx.de (J.M.); lea.fischer@tum.de (L.F.); christoph.schanzenbach@gmx.de (C.S.); fabian.c.schmidt@tum.de (F.C.S.)

**Keywords:** intramembrane proteolysis, substrate recognition, transmembrane domain, BLaTM, heterotypic interaction, γ secretase

## Abstract

Intramembrane proteases, such as γ secretase, typically recruit multiple substrates from an excess of single-span membrane proteins. It is currently unclear to which extent substrate recognition depends on specific interactions of their transmembrane domains (TMDs) with TMDs of a protease. Here, we investigated a large number of potential pairwise interactions between TMDs of γ secretase and a diverse set of its substrates using two different configurations of BLaTM, a genetic reporter system. Our results reveal significant interactions between TMD2 of presenilin, the enzymatic subunit of γ secretase, and the TMD of the amyloid precursor protein, as well as of several other substrates. Presenilin TMD2 is a prime candidate for substrate recruitment, as has been shown from previous studies. In addition, the amyloid precursor protein TMD enters interactions with presenilin TMD 4 as well as with the TMD of nicastrin. Interestingly, the Gly-rich interfaces between the amyloid precursor protein TMD and presenilin TMDs 2 and 4 are highly similar to its homodimerization interface. In terms of methodology, the economics of the newly developed library-based method could prove to be a useful feature in related future work for identifying heterotypic TMD−TMD interactions within other biological contexts.

## 1. Introduction

Sequence-specific interactions between transmembrane domains (TMDs) drive homo- or heterotypic dimerization or oligomerization of many integral membrane proteins in various biological contexts [1,2]. The methods used to investigate TMD−TMD interactions include genetic reporter assays, which translate the interaction of TMDs within a natural cell membrane into reporter gene expressions. Among the various approaches used [3,4,5,6,7,8], the BLaTM system was presented previously [9,10]. The BLaTM system rests on the reconstitution of split β-lactamase by TMD−TMD interactions, which translates into antibiotic resistance of expressing cells. Various configurations of the BLaTM system allow for an analysis of homotypic or heterotypic TMD−TMD interactions. Depending on the transmembrane topologies of the interacting TMDs, their interaction can be parallel [9] or antiparallel [10].

One biological process where TMD−TMD interactions are likely to be relevant corresponds to the recognition of single-span substrate proteins by multi-span intramembrane proteases. Intramembrane proteases cleave peptide bonds within the plane of a lipid bilayer. γ Secretase is arguably the most thoroughly investigated intramembrane protease to date and consists of the catalytic subunit presenilin (PS), which is associated in a 1:1:1:1 stoichiometry with three other subunits, nicastrin, PEN-2, and APH-1 [11,12,13]. γ Secretase has around 150 different known substrates [14]. Therefore, one of the pivotal questions in understanding this protease is how it distinguishes substrates from non-substrates. At a first level of substrate selection, nicastrin appears to serve as a gatekeeper protein whose extracellular domain sterically blocks proteins with large ectodomains that have not been trimmed by shedding [15,16,17]. Nicastrin also recognizes the N-terminus of a substrate’s TMD [18,19,20]. Second, it is thought that a substrate may bind to one or more “exosites” on γ secretase prior to translocating to the catalytic cleft of PS. Early kinetic studies using non-competitive γ secretase inhibitors indeed indicated that an exosite for C99, a fragment produced by shedding of the amyloid precursor protein (APP), is distinct from the catalytic cleft, yet may overlap with it [21]. More recently, γ-secretase subunits that harbor exosites have been identified by photoaffinity mapping using C99 variants with a photocrosslinkable amino acid at various positions. The exosites identified locate to nicastrin, PEN-2, and the PS N-terminal fragment (NTF; containing TMDs 1 through 6), as well as its C-terminal fragment (CTF; TMDs 7 through 9) [22]. While the resolution of these crosslinking studies is limited to the level of individual γ secretase subunits, various biochemical studies have suggested roles of presenilin TMDs 2, 3, 6, and 9 in substrate recognition [23,24]. These TMDs are indeed all located at the periphery of presenilin in the cryoEM structure of γ secretase [18,19,25]. Accordingly, a substrate might translocate from the membrane towards the catalytic site of presenilin through gates formed by TMD2/TM3, TMD2/TMD6, or TMD6/TMD9 pairs, which is supported by various modelling approaches [26,27]. The cryoEM structures [18,19] are believed to represent the substrate/enzyme complexes ready for cleavage as the C-terminal parts of the substrate TMDs are unfolded around their initial cleavage sites and residues downstream of the unfolded parts form a tripartite β-sheet with parts of presenilin. No close contacts between the presenilin TMD2 and a substrate TMD are seen in these structures. Apparently, therefore, the cryoEM structures represent states beyond initial recognition that are ready for cleavage.

To the best of our knowledge, no study has yet systematically investigated the interaction between different substrates and an intramembrane protease at the level of their TMDs. Here, we examined potential pairwise interactions between isolated substrate and γ secretase TMDs in a biological membrane using two different experimental approaches based on the BLaTM system. We identified some pronounced and sequence-specific interactions and discussed the virtues of the different approaches.

## 2. Results

The aim of this study was to perform a systematic survey identifying potential interactions between γ secretase and several of its substrates at the level of their TM helices using BLaTM, a bacterial two-hybrid system [9,10]. The BLaTM system is based on the bacterial co-expression of two separate hybrid proteins, each of which contains a β-lactamase fragment fused to one of the candidate TMDs. An N-terminal β-lactamase fragment (N-BLa) functionally complements the C-terminal fragment (C-BLa) in the *E. coli* periplasm in the event that both fragments are non-covalently linked by interacting TMDs. The strength of the TMD−TMD interaction is reflected by ampicillin LD_50_, the ampicillin concentration permitting the survival of 50% of expressing cells. In BLaTM version 1.2, an N-terminal cleavable signal peptide ensures the N_out_ topology of both hybrid proteins, such that only parallel TMD interactions reconstitute β-lactamase function. Green fluorescent protein (GFP) is fused to the C-terminus of the TMDs and serves to assess the protein expression. Antiparallel TMD−TMD interactions are measured when the N_out_ N-BLa hybrid protein is co-expressed with a C-BLa hybrid (BLaTM 2.0) whose N_in_ topology is ensured by an N-terminal intracellular domain of ToxR. N-BLa 1.2 and C-BLa 2.0 hybrid proteins thus assume antiparallel transmembrane topologies (Appendix A). The BLaTM system has previously been validated using a variety of well-characterized interacting TMDs, including that of glycophorin A (parallel interaction [9]) and EmrE TMD 4 (antiparallel interaction [10]). These TMDs are used as points of reference to which we normalized the strengths of interaction determined here.

The efficiency of β-lactamase complementation does not only depend on the affinity between TMDs, but also on the orientation of the helix−helix interfaces relative to β-lactamase and/or the depth of TMD insertion into the lipid bilayer. Therefore, different orientations or ‘frames’ of candidate TMDs are routinely tested against each other to identify productive pairs [9,10].

### 2.1. Determining TMD−TMD Interactions by Manual Testing of Candidate Pairs

In ‘Approach 1’, employed here (Figure 1a), we identified productive helix−helix pairs by testing four different frames of a given candidate TMD sequence by appending up to three Leu residues to its N-terminus (N_out_ topology) or its C-terminus (N_in_ topology), concurrent with the deletion of up to three natural residues at the opposite terminus (Appendix A). These four different frames were manually tested against four frames of a given partner TMD, yielding a total of 16 combinations for each pair.

As the polytopic presenilin is the enzymatic subunit of γ secretase, we initially focused on interactions between its major isoform PS1 and substrate TMDs. The N-terminus of presenilin faces the cytoplasm; thus, its TMDs 1, 3, 5, 7, and 9 have an N_in_ transmembrane topology, while TMDs 2, 4, 6, and 8 are N_out_, similar to the substrate TMDs (Figure 2). These topologies allow for potential parallel interactions between the TMDs 2, 4, 6, and 8 of PS1 and the APP TMD (Figure 2), which we measured first.

While the LD_50_ values of all 16 combinations of frames are reported for any given TMD pair in Appendix A, the results shown in Figure 3 represent those pairs showing the highest LD_50_ value. A strong heterotypic interaction was detected between APP and PS1 TMD4, where the LD_50_ value strongly exceeded that of the well-known high-affinity GpA homodimer [28]. This interaction was surprising, given that TMD4 was previously not reported to engage in substrate recognition. APP/PS1-TMD2 interactions were tested for different lengths of the TMD2, which was rather long in that it comprised 29 to 32 residues in γ secretase structures [18,19,29]. We reasoned that the integration of this long TMD into the bacterial inner membrane might cause it to assume a tilt angle exceeding its natural angle within presenilin. Excessive tilting might prevent the formation of an extended interface. Indeed, both the shorter version (^125–146^TMD2 and ^133–154^TMD2) produced ~60% to ~70% of the LD_50_ value of GpA, while the long version (^125–154^TMD2) yielded only ~40% of GpA (Figure 3). For all further experiments, ^125–146^TMD2 was thus used. The LD_50_ value of the APP/PS1-TMD8 pair was also of a medium affinity. The APP/PS1-TMD6 pair was similar to the GpA G83I mutant—a low affinity point of reference where a critical Gly of the helix−helix interface was mutated [28].

Potential antiparallel interactions were probed between APP and the N_in_ TMDs 1, 3, 5, 7, and 9 of PS1. We note that residue D385 and K395 of TMD7 and D450 of TMD9 were exchanged for Leu as none of the APP/PS1-TMD7_wt_ or APP/PS1-TMD9_wt_ combinations of frames resulted in detectable ampicillin resistance, suggesting that the wt TMD7 and TMD9 are not properly integrated into the bacterial membrane.

None of the antiparallel APP/PS1-TMD pairs yielded LD_50_ values that reached that of the antiparallel interaction between EmrE TMD4, the positive control. Rather, they were similar to the low-affinity EmrE G90V/G97V (Figure 3 and Appendix A) with its doubly mutated helix−helix interface [10]. Relative LD_50_ values of parallel and antiparallel interactions were difficult to compare to each other as they were normalized to different reference values (GpA vs. EmrE). Absolute LD_50_ values could not be compared either, as the antiparallel configuration of β-lactamase fragments yielded a more efficient functional complementation than the parallel one [10]. Nonetheless, we attempted an approximate comparison of LD_50_ values by relating the parallel and antiparallel LD_50_ values to those of the low-affinity L_20_ TMD. Oligo-Leu helices formed low-affinity membrane-spanning Leu-zippers [30]. Altogether, assuming similar strengths of parallel and antiparallel Leu20 self-interactions yielded a tentative rank order of APP TMD−TMD interactions with PS1 TMD4 >> TMD2 ≈ TMD8 > TMD1 ≈ TMD3 ≈ TMD5 ≈ TMD6 ≈ TMD7 ≈ TMD9 (Figure 3). As the single-span nicastrin and PEN-2 subunits were previously implicated in exosite formation [22]; we also tested the heterotypic interaction of their TMDs with the APP TMD. As shown in Figure 4, a medium-level LD_50_ value was found for the parallel interaction with the N_out_ TMD of nicastrin, while the antiparallel interaction with the N_in_ TMD of PEN-2 was of only a low affinity.

Furthermore, we asked whether the rank order of interaction strength, as seen here with γ secretase/APP TMD−TMD interactions, was preserved with other well-characterized substrates, namely Notch1, N-cadherin, and ErbB4. In previous cleavage assays, K_M_ values that approximated the substrate/enzyme affinity had been found for APP and these substrates were found to be similar within an order of magnitude (APP, K_M_ = 0.4 µM; Notch1, 1.08 µM; N-cadherin, 1.46 µM; ErbB4, 0.31 µM) [31]. Again, 16 combinations of TMD frames (Appendix A) were measured for each TMD pair in order to sample the potential geometries of helix−helix interactions as thoroughly as in the case of APP (Appendix A). To our surprise, however, the LD_50_ values obtained with these substrate TMDs in combination with the TMDs of PS1 (Figure 3), nicastrin, or PEN-2 (Figure 4) were of a uniformly low affinity with only minor variations. We also included the TMD of the β1 subunit of human integrin (ITGB1), one of the few firmly established non-substrates of γ secretase [32]. By implication, its interactions with γ-secretase TMDs were not believed to be of functional importance. As expected, only low-affinity interactions were detected, except for the case of the ITGB1/PS1-TMD8 pair with its medium-level LD_50_.

To exclude that the different LD_50_ values measured here originated from different expression levels of BLa hybrid proteins, we routinely monitored the fluorescence of the BLa/GFP hybrid proteins in the cells. GFP fluorescence varied only slightly between most constructs (Appendix A). As those variations did not coincide with the much larger variations in LD_50_, the latter did not arise from different expression levels. In another control, we tested the potential impact of homotypic interactions. A heterotypic TMD−TMD interaction was in equilibrium with the potential homotypic interactions of the partner TMDs. Thus, a strong homotypic interaction might compete with the heterotypic interaction and thus indirectly lower the LD_50_ values attributed to the latter. As shown in Appendix A (γ-secretase TMDs) and Appendix A (substrate TMDs), most homotypic interactions were of only a modest affinity; however, with LD_50_ values remained below 50% of GpA. Medium-level self-interaction was seen with frames 1 and 3 of N-cadherin and frame 1 of ITGB1 (Appendix A). Importantly, the homotypic interactions did not anti-correlate with the strengths of the corresponding heterotypic interactions given in Figure 3 and Figure 4, thus ruling out a significant effect of homotypic interaction.

In order to examine the sequence specificity of APP/PS1 TMD2 and TMD4 interactions, we subjected the APP TMD to Ala/Ile-scanning mutagenesis and determined the resulting LD_50_ values. In the case of APP/PS1 TMD4, mutating 9/22 residues reduced the LD_50_ by >50% of wt. These nine critical residues were likely to contribute to the helix−helix interface (Figure 5a), which could also include the mutation-sensitive I32, I41, T43, and V44. These 9 critical residues also proved to be the most sensitive ones when testing helix-helix interaction with PS1 ^125–146^TMD2 (Figure 5b). The most sensitive residues corresponded to Gly and Ala. Interestingly, the pattern of critical residues suggested that they formed two separate interfacial motifs with helical periodicity: G29, A30, G33, G37, V40, T48 vs. M35, G38, and A42 (highlighted by different coloring). Figure 5d depicts these motifs on a helical wheel plot and on the NMR structure of the APP TMD [33], and reveals that both motifs located to opposite faces of the helix. We also note that the highly mutation-sensitive G37 and G38 were part of the G_37_G_38_ motif forming a hinge in the TMD [33,34]. Again, expression levels were controlled for by monitoring the GFP fluorescence of the cells, which confirmed that the impact of the mutations did not result from a reduced protein expression (Appendix A). The APP TMD mutants were also tested for homotypic interaction under the same conditions. Although the homotypic signal was quite low (Appendix A), we detected a number of mutation-sensitive residues; most of these corresponded to the interfacial residues participating in the heterotypic interfaces described above.

Overall, our results revealed strong and sequence-specific interactions between the APP TMD and TMDs 2 and 4 of PS1, as well as the nicastrin TMD. On the one hand, this was consistent with a role of TMD−TMD interactions in APP recognition by γ secretase. On the other hand, none of the other well-established substrate TMDs tested here interacted more efficiently with γ-secretase than the TMD of the non-substrate ITGB1, which questions the general role of TMD−TMD interactions in substrate recruitment by γ-secretase.

### 2.2. TMD−TMD Interactions Determined by Library Screening

Given that presenilin TMD2 has been frequently been implied in substrate recognition and gating [23,24,26,35,36], the observation that we could not detect strong interactions between PS1 TMD2 and substrates beyond APP prompted the question of whether or not the exceedingly long PS1 TMD2 was represented well enough by the four frames of ^125–146^TMD2 for the heterotypic assembly with the non-APP substrate TMDs. In ‘Approach 2′, we thus expanded the analysis by testing additional frames of the presenilin TMD2. To cover the complete length of TMD2, we tested a total of 12 frames, including frames 0–3 tested in Approach 1 (Appendix A), using an economical library-based method. Each frame contained 22 residues, similar to the 21 to 24 residues of the substrate TMDs.

In the library screen, a corresponding set of 12 TMD2-encoding C-BLa plasmids, N-BLa was co-expressed with a plasmid pool encoding a total of 28 TMDs (four frames each). In addition to the APP TMD, the pool comprised the TMDs of 26 additional well-established γ secretase substrates and the non-substrate integrin β1 [14,37] (Appendix A), resulting in 1344 individually screened interactions. Transformed *E. coli* cells were grown in 12 different ampicillin concentrations in liquid LB medium. The abundance of clones under these selective conditions, as determined by the next-generation-sequencing (NGS) of the reading frames, reflected ampicillin resistance, i.e., the strength of interaction. This approach was initially tested by comparing the abundances of clones corresponding to a set of reference TMDs to their respective LD_50_ values obtained by manual analysis. Indeed, both data sets correlated very well (R^2^ = 0.94, Appendix A), thus validating Approach 2.

The results of the screen for new interacting pairs are shown by a heatmap representing the abundances of all of the tested pairs (Appendix A), while Figure 6 compares only the pairs whose signals exceeded 40% of the GpA reference. Accordingly, APP frame 2 paired with frames -3 and -1 of presenilin TMD2 even more efficiently than with frame 0, whose LD_50_ was also covered by approach 1 (Figure 3). The screen also revealed a number of promising new candidates. Specifically, PS1 TMD2 frame -3 also scored with the CD44 TMD (frame 2), while TMD2 frames 7 and 8 appeared to interact with TMDs from a diversity of substrates. For example, these pairs included members of the Notch receptor family (Figure 6 and Appendix A), excluding Notch1 that also failed to strongly interact with the PS1 TMD2 variants tested in Approach 1 (Figure 3). Unexpectedly, a strong signal was also seen with the presumed non-substrate ITB1 TMD.

Overall, library-based Approach 2 proved to be an economical way to exhaustively screen for new TMDs that may interact with presenilin TMD2 during substrate recognition.

## 3. Discussion

Our results shed light on the potential role of pairwise TMD−TMD interactions in substrate recruitment by γ secretase, yet hint that different substrates might behave differently. On the one hand, the data indicate that APP TMD binds with efficiencies well above that seen for the non-substrate ITGB1 to the isolated TMD2 and especially to the TMD4 of presenilin, as well as to the TMD of nicastrin. That nicastrin participates in substrate selection is well known. This subunit has been ascribed a gate keeper role whose function is to repel proteins with large ectodomains [17] and/or to actively bind the N-terminus of C99 [20]. In addition, a previous photocrosslinking approach indicated the close proximity of nicastrin and the N-terminal juxtamembrane domain of C99 at an early stage of substrate recognition [22]. Interactions between the TMDs of both proteins have not yet been described, yet may contribute to C99 recruitment, as suggested by our present data. After contacting nicastrin, C99 sequentially binds to exosites located in the NTF and CTF of presenilin [22]. At the level of individual TMDs, exchanging either TMD2 or part of TMD6 of presenilin for non-related TMDs had previously abolished labeling by a substrate-based photoprobe, thus implying TMD2 and the luminal side of TMD6 in substrate binding [35]. In addition to presenilin TMD2 and TMD6 [35,36], TMD9 [38] has been implicated in substrate binding by a range of other biochemical studies, including Cys labelling and substrate crosslinking. Various molecular dynamics studies support the contribution of presenilin TMD2 and TMD6 to C99 recognition and/or gate formation [26,39,40,41,42]. Our current finding that the isolated APP TMD binds to the PS1 TMD2 with remarkable strength is thus in line with previous studies involving the complete γ secretase protein in eukaryotic membranes. A surprising observation made here is that TMD4 is the PS1 TMD that interacts most strongly with APP. Indeed, the APP/PS1-TMD4 interaction is the strongest heterotypic interaction that has been observed so far using the BLaTM system [9,10]. Although PS1 TMD4 had hitherto not been proposed to function in substrate recognition, circumstantial evidence may support such a function. For example, mutating some of its residues (I196, L201, and V207) had previously reduced Aβ40 and Aβ42 production while mutating others (A199, I202, G206, V208, G209, and M210) changed the Aβ40/Aβ42 ratio [43]. In addition, TMD4 harbors a number of mutations associated with familial Alzheimer’s disease (FAD) (www.alzforum.org, accessed on 15 December 2022). Specifically, G206 and G209 residues appear to be hot spots of mutation. In FAD, G206 is mutated to Ala, Asp, Ser, and Val, while G209 is exchanged for Ala, Glu, Arg, and Val. Furthermore, TMD4 was previously crosslinked to TMD7 after introducing appropriate pairs of Cys residues. Specifically, a Cys at position 213 near the TMD4 C-terminus can be crosslinked to Cys at positions 383 or 387 near the TMD7 N-terminus [43]. Thus, both TMDs directly contact each other at least transiently near D385, one of the catalytic amino acids, although the respective C_α_-C_α_ distances are 1.3 nm (I213-L383) and 1.7 nm (I213-I387) in the C83/γ secretase complex [18] at cryogenic temperatures. This transient TMD4/TMD7 contact appears to require a conformationally flexible TMD4 as the G206L mutation abolished crosslinking [43]. Gly residues are known to enhance the conformational flexibility of a TM helix [44]. These considerations raise the possibility of an alternative pathway of substrate entry where TMD4 might play a role in recognizing C99 and transferring it towards the catalytic cleft.

The nature of APP/presenilin TMD interfaces identified by Approach 1 was investigated by scanning mutagenesis. The results identified interfacial residues including GxxxG motifs that are known to contribute to TM helix−helix interactions [45]. Both the G29xxxG33 and G38xxxA42 motifs had previously been implied in forming alternative interfaces of APP TMD−TMD homodimers [46,47,48,49,50]. Our present data show that these motifs also support heterotypic interactions of the APP TMD. Specifically, the G29xxxG33xxxG37 or G38xxxA42 motifs identified here might form alternative interfaces at opposite faces of the TM helix that control heteromerization with PS1 TMD2 and TMD4. As a homodimeric C99 fragment of APP is not cleaved by γ secretase [51], one may speculate that residues forming a homodimerization interface contact presenilin TMDs after substrate monomerization.

Interestingly, testing different parts of TMD2 in the library-based approach 2 uncovered alternative modes of interaction with the APP TMD as well as interactions with the TMDs of several other γ secretase substrates. These other substrate TMDs were preferably bound to frames -7 and -8 of PS1 TMD2. It appears, therefore, that different faces of presenilin TMD2 may be able to recognize different substrate TMDs. Frames -7 and -8 include a C-terminal Tyr; possibly, this Tyr stabilizes the TMD2 helix in the bacterial membrane [52], thus increasing its potential to bind partner TMDs. Extending the search for alternative substrate-binding sites to other presenilin TMDs will be the subject of future work.

On the other hand, most substrate TMDs tested here did not interact strongly with presenilin TMD2. This may be for a number of reasons and points to some potential limitations of testing interactions between pairs of isolated TMDs. First, we speculate that the TMD of APP might represent a special case where its binding to PS1 TMDs 2 and 4, as well as to the nicastrin TMD may be more relevant for the recognition of C99 than of other substrates by γ secretase. Second, we may not have captured some heterotypic TM helix-helix interfaces in the bacterial membrane that may exist in the eukaryotic cell. Possible reasons for missing such interfaces include (i) different depths of insertion of both isolated partner TMDs in the membrane, thus leading to cognate helix surfaces being out of register and (ii) unnatural tilt angles assumed by the helices that prohibit the formation of extended helix−helix interfaces, among others. Third, potentially strong homotypic substrate TMD−TMD interactions might compete for heterotypic interactions in the BLaTM system. However, only in the case of N-cadherin did we detect medium-level self-interaction. N-cadherin TMD homomerization was on par with the APP/PS1-TMD2 interaction and might thus compete with heterotypic N-cadherin/γ secretase TMD−TMD binding.

## 4. Materials and Methods

### 4.1. Plasmid Design and Construction

Plasmids were constructed as previously described [9,10]. Briefly, low-copy plasmids encode either the N-BLa or the C-BLa hybrid protein. The origins of replication were compatible with maintaining both plasmids within one bacterial cell. Protein expression was induced by activation of the arabinose promoter pBAD of the low-copy plasmids. In BLaTM 1.2, the N-BLa and C-BLa plasmid encoded the N-terminal or C-terminal part of split β-lactamase, respectively, whose N-termini were fused to cleavable signal peptides to ensure their N_out_ topology. The TMDs of interest were linked to the BLa-protein via flexible GGS linkers and on the C-termini of the TMDs; a rigid helical linker connected it to superfolder GFP. In BLaTM 2.0, the N-BLa plasmid of BLaTM 1.2 was combined with a C-BLa plasmid where the TMD was connected via a flexible spacer to the C-terminus of the cytoplasmic domain of the ToxR protein and to the N-terminus of the C-BLa fragment [10]. For the insertion of TMD-encoding sequences into a plasmid, corresponding oligonucleotides were hybridized. The resulting cassette was phosphorylated and ligated into previously digested vectors (using NheI and BamHI). The sequences of all γ secretase TMDs corresponded to the TMD helices annotated in the cryoEM structure (pdb 5fn2 [29]). To achieve different orientations of the TM helices within the membrane, four different frames were designed in each case by appending Leu residues to N-termini and C-termini, respectively, as described in the main text (Appendix A).

### 4.2. Determining Ampicillin LD_50_ Values

Ampicillin LD_50_ determination was done as described previously [9]. Briefly, cotransformation of competent *E. coli* JM83 cells with N-BLa and C-BLa plasmids (each encoding one TMD of interest) was performed followed by overnight incubation at 37 °C on LB-agar plates containing chloramphenicol (Cm; 34 µg/mL) plus kanamycin (Kan; 35 µg/mL) to ensure plasmid inheritance. Afterwards, 10 colonies were incubated in 5 mL LB medium (Cm, Kan) in a turning wheel at 37 °C overnight. Next, 4.5 mL of expression medium (Cm, Kan, 1.33 mM arabinose, 0.3 mM isopropyl-thio-galactopyranoside (IPTG) for BLaTM 1.2 experiments or 0.6 mM IPTG for BLaTM 2.0 experiments) was inoculated with 500 µL of the overnight culture. During incubation, a fresh stock solution of ampicillin (20 mg/mL) was prepared. Subsequently, 1 mL of expression medium containing ampicillin ranging from 0 to 1000 µg/mL (depending on the investigated TMDs) was added to each well of a 12-well plate (Greiner Bio- one). The plates were sealed and prewarmed until further use. After incubation of the expression culture for 4 h at 37 °C, the culture was diluted to an OD_600_ = 0.2. Then, 1 mL of the diluted culture was given to each well of the 12-well plates, resulting in an OD_600_ = 0.1. The plates were sealed in a moisturized container and incubated at 37 °C and 140 rpm on a shaker for 19 h (shaking amplitude 12 mm, AK82, Infors AG). The following day, extinction at λ = 544 nm was measured with a microplate reader (FluoStar, BMG Labtech, Ortenberg, Germany). After fitting the data points with the Hill equation using ECCpy (an open source program in python (https://github.com/teese/eccpy, accessed on 15 December 2022 [9]), the LD_50_ values were calculated.

### 4.3. Determining GFP Expression

The GFP expression was determined in parallel with the LD_50_ measurements using aliquots of the expression cultures that had been incubated for 4 h at 37 °C. The cell density at 600 nm was measured to calibrate fluorescence measurements to the cell count and 200 µL of the 4 h cultures were centrifuged at 19.000× *g* for 1 min. Pellets were resuspended in 200 µL phosphate buffered saline (PBS) and transferred to a black Nunc 96-well plate to measure GFP fluorescence (λ_Ex_ = 485 nm, λ_Em_ = 520 nm, FluoStar, BMG Labtech). Uninduced cells were measured as well and the values were subtracted from the measured data. The fluorescence signal per single cell was calculated using the assumption that an OD_600_ = 1.0 equals 8 × 10^8^ cells/mL.

### 4.4. Design and Screening of Combinatorial TMD Libraries

N-BLa and C-BLa plasmids expressing reference TMDs were individually co-transformed into *E. coli* JM83 and plated onto LB Cm (34 µg/mL)/Kan (35 µg/mL) agar. Pools of oligonucleotides encoding the TMDs of the combinatorial libraries were synthesized (Twist Bioscience, South San Francisco, CA, USA) and amplified according to the manufacturer’s recommendations using 5′ biotinylated oligonucleotides (Eurofins Genomics, Ebersberg, Germany) given in Appendix A. The N-BLa 1.2 vector was digested with NheI and BamHI, dephosphorylated with alkaline phosphatase (Thermo Fisher Scientific, Waltham, MA, USA), and purified using spin-columns (NucleoSpin Gel and PCR Clean-up, Machery-Nagel, Dueren, Germany). The biotinylated, amplified pool was also digested using NheI and BamHI. Digested fragments were subsequently purified from undigested DNA fragments with streptavidin-coated magnetic beads (Merck Millipore, Burlington, MA, USA). Ligation was performed with T4-DNA ligase (Thermo Fisher Scientific, Waltham, MA, USA) according to the manufacturer’s recommendations using 1 µg vector and a vector−insert ratio of 1:5. *E. coli* JM83 was transformed with our N-BLa 1.2 pool and competent cells were generated from the transformants (referred to as JM83 pool). C-BLa 1.2 constructs containing PS1-TMD2 variants (C-BLa1.2_TMD2_-3 through C-BLa1.2_TMD2_8) were cloned individually between NheI and BamHI sites of C-Bla1.2 and the sequences were verified via sanger sequencing (Eurofins Genomics, Ebersberg, Germany). These C-BLa 1.2 constructs were then used to transform 12 individual aliquots of the JM83 pool using 1 µg of DNA for each transformation. Then, 5% of the recovered clones were plated onto LB Cm/Kan agar to calculate the transformation efficiency, while 50 mL of liquid LB Kan/Cm medium was inoculated with the remaining 95%. The controls were prepared as previously described in Approach 1 and the pool transformations were incubated overnight at 37 °C in a turning wheel incubator.

A total of 16 aliquots of induction medium (4.5 mL containing LB Cm/Kan, 1.33 mM arabinose, and 0.3 mM isopropyl-thio-galactopyranoside (IPTG)) were inoculated with either 0.5 mL of the densely grown overnight cultures transformed with the four controls or one of the 12 pools and incubated at 37 °C for exactly 4 h. A cell density at 600 nm was measured and the pools and controls were mixed at an appropriate ratio (pool:GpA wt:GpA G83I:APP-TMD2:APP-TMD4 = 112:2:2:2:2). Test tubes containing induction medium plus an ampicillin gradient ranging from 0 to 32 µg/mL in 4 µg/mL increments were inoculated to an OD600 = 0.1 and incubated in a turning wheel incubator at 37 °C for exactly 15 h. Plasmids of the surviving clones were subsequently isolated by spin column purification. Prior to NGS, the aliquot containing the ampicillin concentration ensuring an optimal dynamic range in selection was assessed by qPCR (as different batches of ampicillin may vary in antimicrobial efficiency due to ampicillin breakdown). For the primers used, refer to Appendix A. During qPCR, the ratio of the abundances seen for the surviving N-BLa 1.2 GpA wt and GpA G83I plasmids was determined. For the actual selection experiment, we then used the ampicillin concentration differentiating most efficiently between both reference TMDs, where the GpA wt:GpA G83I ratio ranged in between 4:1 and 5:1. After each of the 12 screenings, the TMD-coding sequences in our input pool (IN, 0 µg/mL Amp) and the best-condition pool (OUT, between 16 and 28 µg/mL Amp, as determined via qPCR) were amplified via PCR carrying universal Illumina 5′ sequencing adapters (Appendix A). The resulting amplicon were sequenced at Eurofins Genomics, Germany (NGSelect Amplicon 2nd PCR). The NGS results were evaluated using a custom python script. Briefly, each read was mapped to the known pool member sequences, their abundance was quantified, and the OUT/IN ratios of all read counts were calculated and normalized to the GpA wt levels.

## 5. Conclusions

We conclude that in the case of APP, sequence-specific substrate/enzyme interactions can be detected at the level of the TMDs, which may originate from the unusual sequence of the APP TMD with its sticky GxxxG motifs. Several other substrate TMDs also interact with different constructs of PS1 TMD2. γ Secretase cleaves about 150 different substrates [14]. Given that their TMDs do not share much sequence homology, one may wonder whether it is reasonable to expect sequence-specific interactions between all of them and TMD2. Rather, other substrates might translocate via different pathways to the catalytic site of presenilin [26]. Future experiments will reveal the extent of the pairwise interactions between γ secretase and the full complement of substrate TMDs. We envision that the library-based approach 2 explored here will be instrumental in such analyses. The advantage of the library-based method is the drastically reduced effort when compared with manual LD_50_ determination. Approach 2 is thus a promising way of economically exploring a multitude of potential heterotypic TMD interactions. Apart from substrate enzyme interactions in intramembrane proteolysis, the efficient detection of heterotypic TMD−TMD interactions may also prove useful in other biological contexts [53,54,55].

In sum, the strength of the BLaTM-based method is that a multitude of different TMD pairs can be tested for potential interactions, in particular when employing approach 2. Limitations may arise from potential false-negative results due to non-aligned TMD interfaces in the membrane, strongly tilted TMDs, and competition of heterotypic by strong homotypic interactions.

## Figures and Tables

**Figure 1 ijms-24-14396-f001:**
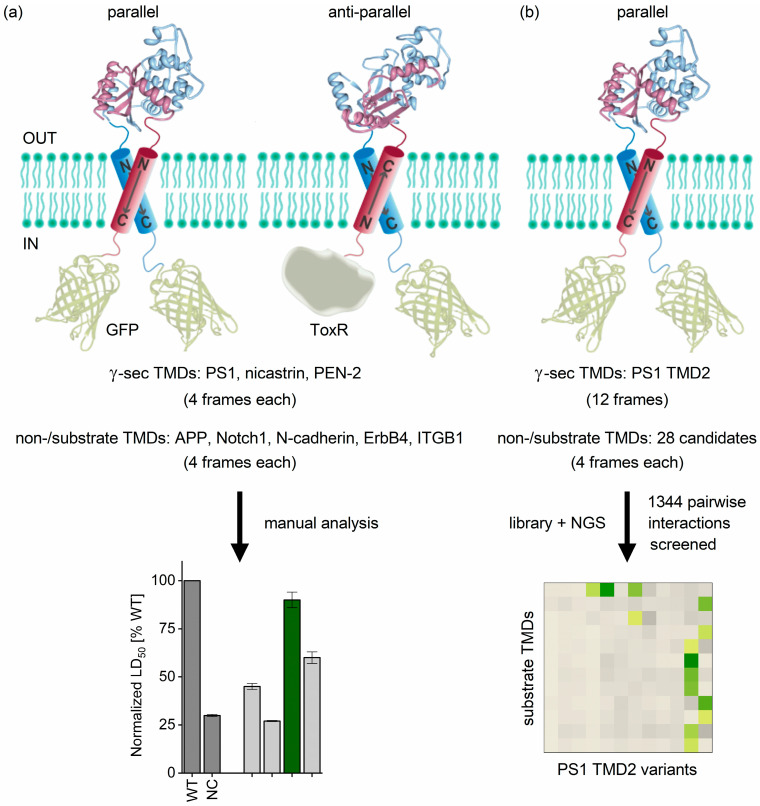
Comparing both approaches used here to investigate substrate/γ secretase TMD−TMD interactions. (**a**) In Approach 1, we manually tested four substrate TMDs against most γ secretase TMDs. LD_50_ values represent relative affinities. (**b**) In Approach 2, 28 substrate TMDs were run simultaneously against presenilin TMD2, which corresponds to the γ secretase TMD implied by other studies in substrate recognition, using a highly efficient screening technique. Sequence abundance under selective conditions is equivalent to affinity and encoded by the colors of the heatmap (yellow: lowest affinity; dark green: highest affinity). WT = wild type; NC = negative control. Soluble domains extending into the bacterial periplasmic space represent β-lactamase domains; GFP and ToxR domains pointing to the cytoplasm are annotated.

**Figure 2 ijms-24-14396-f002:**
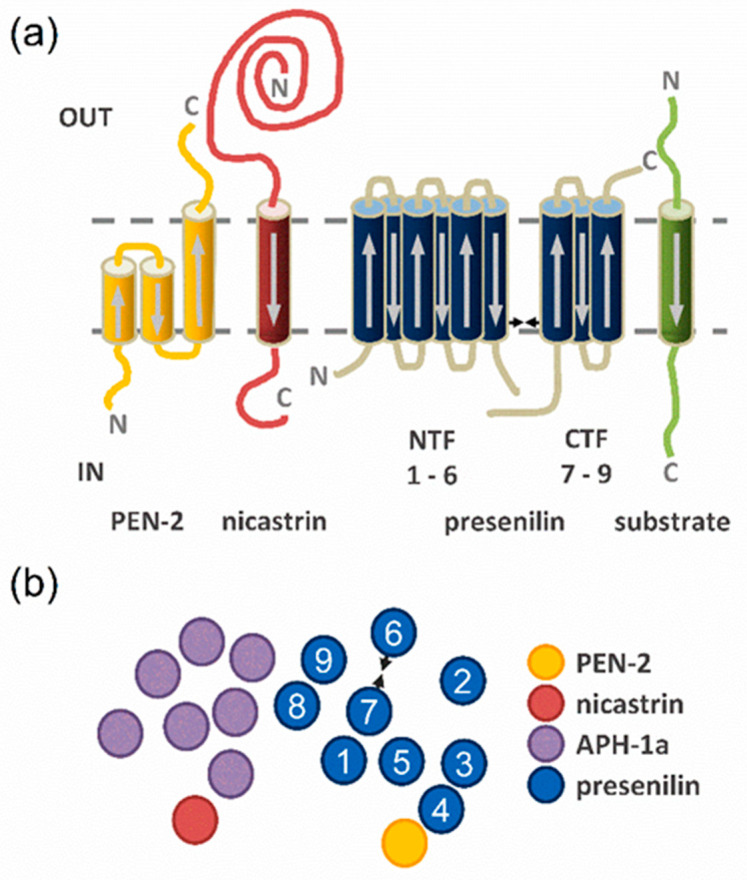
Overview of γ-secretase and substrate TMDs. (**a**) Transmembrane topologies of the γ-secretase subunits presenilin (blue), PEN-2 (yellow), nicastrin (red), APH-1 (purple), and a substrate (green). Arrows correspond to the direction of the sequences. (**b**) Top view onto the γ-secretase TMDs. The two catalytic aspartates in TMD6 and TMD7 of presenilin are represented by arrowheads and the TMDs are numbered from N- to C-terminus.

**Figure 3 ijms-24-14396-f003:**
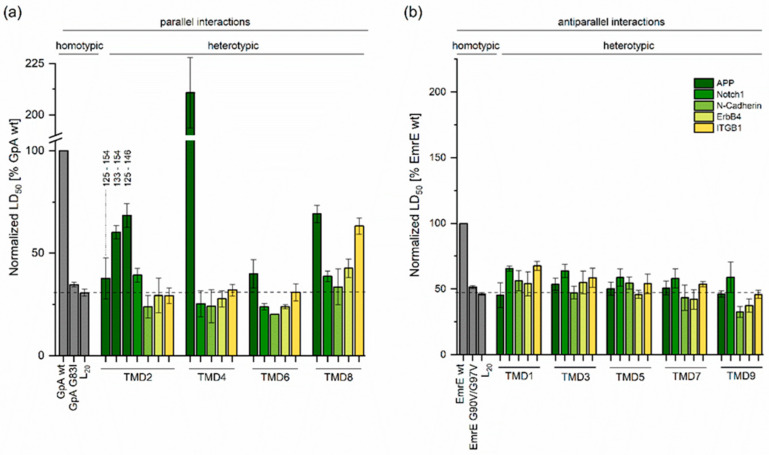
TMD−TMD interactions of PS1 with substrates and the non-substrate ITGB1. (**a**) Strength of parallel heterotypic interactions normalized to the homodimerization signal of GpA used as a reference. Three different variants of PS1 TMD2 were tested in combination with APP. (**b**) Antiparallel heterotypic interactions normalized to the homodimerization signal of EmrE. The shown data correspond to the combination of TMD frames showing the strongest interactions in any given case (see Appendix A A-T, where the GFP expression controls are also given). Means ± SEM, n > 3. The positive and negative controls (grey bars) were included in every single round of experiments.

**Figure 4 ijms-24-14396-f004:**
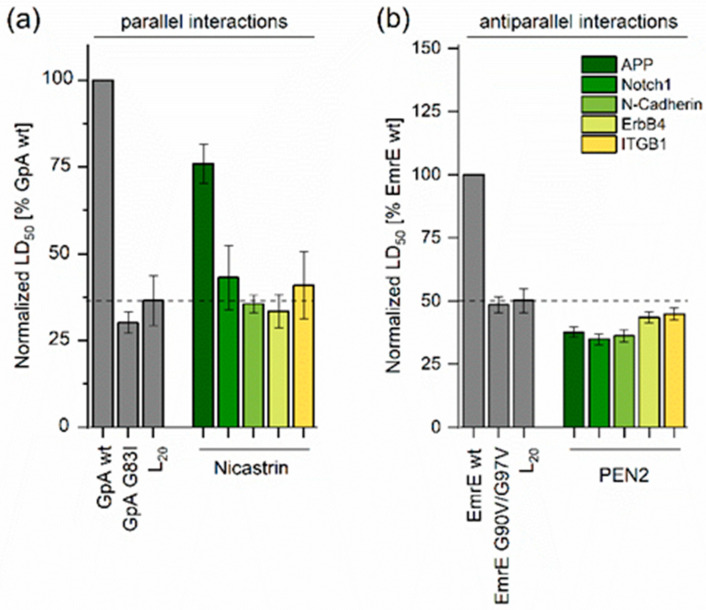
TMD−TMD interactions of nicastrin or PEN-2 with substrates and the non-substrate ITGB1. (**a**) Strength of parallel heterotypic interactions to the nicastrin TMD normalized to the homodimerization signal of GpA. (**b**) Strength of antiparallel heterotypic interactions to the PEN-2 TMD normalized to the homodimerization signal of EmrE TMD4. The shown data correspond to the TMD pairs showing the strongest interactions, as shown in Appendix A, which also contains the GFP expression controls. Means ± SEM, n > 3. The positive and negative controls (grey bars) were included in every single round of experiments. The color coding given in the inset of panel (**b**) also applies to panel (**a**).

**Figure 5 ijms-24-14396-f005:**
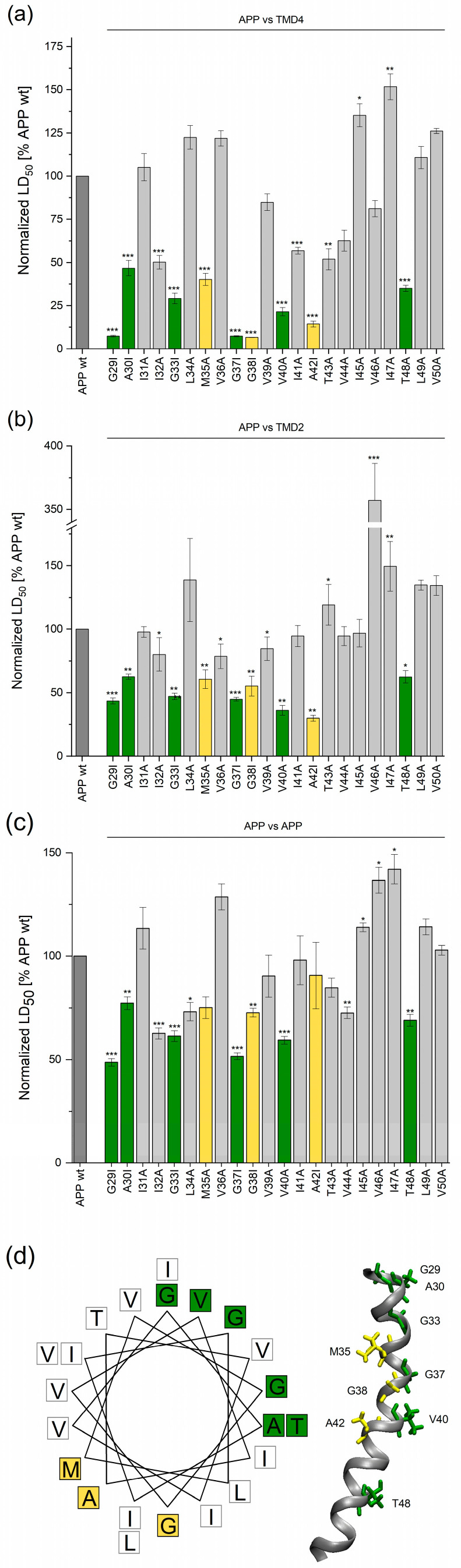
Mutational analysis of the APP TMD in its pairwise interactions with different PS1 TMDs or itself. (**a**) Strength of the interaction of APP TMD (frame 2) mutants vs. PS1 TMD4 (frame 3) normalized to the signal of wt APP TMD. Residues whose mutation reduced the signal to <50% of wt are colored. (**b**) Strength of interaction of APP TMD (frame 2) mutants vs. PS1 ^125–146^TMD2 (frame 0) normalized to the signal of wt. We note that mutant V46A more than tripled the LD_50_ in this case, for reasons that are unclear. For technical reasons, a lower LD_50_ limits the potential impact of mutations. (**c**) Strength of homodimerization of APP TMD (frame 2) mutants normalized to the signal of the wt APP TMD. The same residues as in (**a**) are highlighted in (**b**,**c**). (**d**) Mapping the mutation-sensitive residue positions onto a helical wheel or the NMR structure of the helix (pdb: 6hyf) model suggests that the amino acids colored in yellow or green, respectively, may correspond to two separate helix−helix interfaces formed by the APP TMD. Single, double, or triple asterisks denote statistical significance at the 0.05, 0.01, or 0.001 confidence levels (relative to wt APP). Means ± SEM, n = 4–5. The pairs used as references (dark grey bars) were included in every single round of the respective experiments.

**Figure 6 ijms-24-14396-f006:**
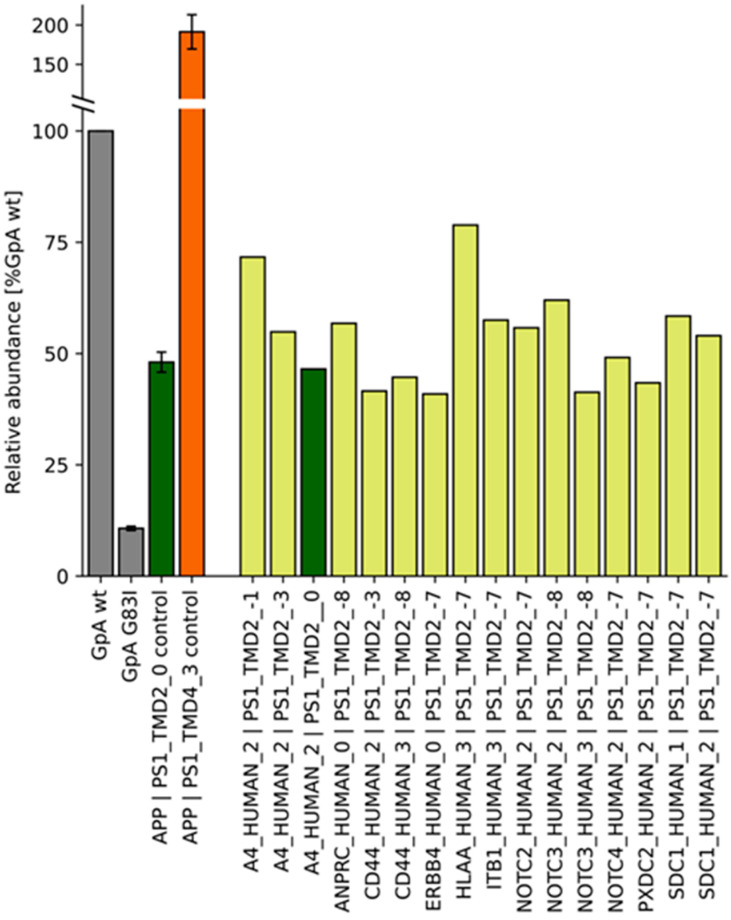
TMD−TMD interactions of various parts of PS1 TMD2 with substrate TMDs, as identified by the BLaTM library screening approach. Candidate pairs (shown in yellow) were identified by next generation sequencing and the resulting full dataset of 1344 pairs (see Appendix A) was filtered for abundances exceeding 40% of the signal of GpA wt. Homotypic interactions of positive and negative controls are given by grey bars; heterotypic TMD pairs also covered by approach 1 (Figure 3) are shown in dark green and orange. The TMD sequences of the pair A4_HUMAN_2|PS1_TMD2_0 (identified by approach 2) are equivalent to those of the APP | PS1_TMD2_0 pair (approach 1) (both colored in green).

## Data Availability

Data is contained within the article or Appendix A.

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
