# Peer review of "Interaction of Substrates with γ-Secretase at the Level of Individual Transmembrane Helices—A Methodological Approach"

_ijms, 2023, doi:10.3390/ijms241814396_

Round 1
Reviewer 1 Report
Interaction of Substrates with gamma-Secretase at the Level of Individual Transmembrane Helices – a Methodological Approach
Theresa M. Pauli et al.
The paper describes the measurement of interactions of isolated transmembrane domains using a beta-lactamase complementation approach. The readout is the LD50 for ampicillin of transformed E.coli. As far as I can judge, the methods have been used correctly and the authors present a large number of measurements. The overall message of the paper is clear, but I found the paper not easy to read due to the many details and the way they were presented.
What is missing in my eyes is a discussion of the correlation with some recent cryoEM structures of gamma-Secretase in complex with Notch (6idf) and APP (6iyc). Things that could be discussed are e.g.
- Does the shorter fragment of TDM2 (lines 137-139) correspond to the interface observed in the cryoEM structures?
- Do the APP residues identified in the ALA scanning experiment (Figure 5) also interact with PS1 in the cryoEM structure? Which residues do, and which ones don’t?
- In the discussion (lines 322-323) the interaction between TMD4 and APP is called surprising, suggesting that this interaction is not observed in the cryoEM structure. Is this correct? In general, why is this observation surprising?
Here a good correlation would point to reliable results. However, a poor correlation could point to dynamic changes during the binding process, as is already discussed by the authors in the discussion section, based on mutations. This could be extended to the cryoEM data.
Some of the result do not conform to literature: e.g. with PS1, nicastrin and PEN-2 (lines 191-194) and with the presumed non-substrate ITB1 TMD (lines 283-284). The authors should mention this also in the discussion as a cautionary note.
Some minor points:
Figure 1: This figure is much poorer annotated as the similar figure S1. Here the authors should add labels for GFP, ToxR, IN and OUT similar to figure S2. In the legend it might be good to mention: WT=wild type, NC=negative control, if this is what is meant be the abbreviations. By the colors of the heat map, one should add some explanation to the colors, e.g. green: highest activity.
Line 134: “TMD2, which is exceptionally long”. According to ref. 19, TDM2 is about as long as TDM1 and shorter as TDM3. I would remove the word “exceptionally”
Figure 3 plus text: Why were for the parallel interaction only TMD’s 2, 4, 6 and 8 tested and for the antiparallel interactions only TMD’s 1, 3, 5, 7 and 9? Has this to do with the orientation of the TMD’s in the full-length protein? This should be mentioned somewhere.
Line 253: section “2.” should become “2.2”.
Author Response
Reviewer 1
The paper describes the measurement of interactions of isolated transmembrane domains using a beta-lactamase complementation approach. The readout is the LD50 for ampicillin of transformed E.coli. As far as I can judge, the methods have been used correctly and the authors present a large number of measurements. The overall message of the paper is clear, but I found the paper not easy to read due to the many details and the way they were presented.
What is missing in my eyes is a discussion of the correlation with some recent cryoEM structures of gamma-Secretase in complex with Notch (6idf) and APP (6iyc). Things that could be discussed are e.g.
We are grateful to the reviewer for carefully reviewing our paper and for bringing up some useful points to improve the manuscript.
- Does the shorter fragment of TDM2 (lines 137-139) correspond to the interface observed in the cryoEM structures?
Response: Actually, there is no substrate/enzyme helix/helix interface in the cryoEM structures of refs 18 and 19. Rather, the C-terminal part of the TMDs of Notch or APP contact residues at the PS1 TMD6 C-terminus and residues within the loop linking TMDs 6 and 7 (forming a tripartite sheet structure). The N-terminal parts of both substrate TMDs are mostly in contact with lipids. Thus, the structures are considered to represent states after initial recognition. The evidence for a role of PS1 TMD2 in substrate recognition is based on biochemical experiments (cited in our paper).
- Do the APP residues identified in the ALA scanning experiment (Figure 5) also interact with PS1 in the cryoEM structure? Which residues do, and which ones don’t?
Response: As outlined above, the cryoEM structure does not show interfaces between the helical parts of the substrate TMDs and presenilin.
- In the discussion (lines 322-323) the interaction between TMD4 and APP is called surprising, suggesting that this interaction is not observed in the cryoEM structure. Is this correct? In general, why is this observation surprising?
Response: Again, no interface to TMD4 is seen in the structure. The reason why we believe that our PS1 TMD4 / APP TMD interaction is surprising is the lack of biochemical evidence for this interaction in the literature. We have now added this to the manuscript.
Here a good correlation would point to reliable results. However, a poor correlation could point to dynamic changes during the binding process, as is already discussed by the authors in the discussion section, based on mutations. This could be extended to the cryoEM data.
Response: Indeed, the interaction is a dynamic one and no 3D structure exists showing its initial states. This is now made more explicit in the introduction of the manuscript.
Some of the result do not conform to literature: e.g. with PS1, nicastrin and PEN-2 (lines 191-194) and with the presumed non-substrate ITB1 TMD (lines 283-284). The authors should mention this also in the discussion as a cautionary note.
Response: Good idea, we make it explicit in the revised manuscript which interactions of the APP TMD with g-secretase are indeed stronger than those seen with the non-substrate ITGB4 (whose interactions with g-secretase TMDs are by implication not believed to be of functional importance).
Some minor points:
Figure 1: This figure is much poorer annotated as the similar figure S1. Here the authors should add labels for GFP, ToxR, IN and OUT similar to figure S2. In the legend it might be good to mention: WT=wild type, NC=negative control, if this is what is meant be the abbreviations. By the colors of the heat map, one should add some explanation to the colors, e.g. green: highest activity.
Response: Good idea – done!
Line 134: “TMD2, which is exceptionally long”. According to ref. 19, TDM2 is about as long as TDM1 and shorter as TDM3. I would remove the word “exceptionally”
Response: We replaced “exceptionally” by “rather”.
Figure 3 plus text: Why were for the parallel interaction only TMD’s 2, 4, 6 and 8 tested and for the antiparallel interactions only TMD’s 1, 3, 5, 7 and 9? Has this to do with the orientation of the TMD’s in the full-length protein? This should be mentioned somewhere.
Response: Absolutely, only parallel interactions are conceivable between the Nout TMDs of the substrates and the Nout TMDs 2,4,6 and 8 of PS while antiparallel interactions are possible to PS TMDs 1,3,5,7,9. We make this more explicit in the revised Results.!
Line 253: section “2.” should become “2.2”
Response: done!

Reviewer 2 Report
No study has yet systematically investigated the interaction between different substrates and an intramembrane protease at the level of their TMDs. Here, authors examined potential pairwise interactions between isolated substrate and γ−secretase TMDs in a biological membrane using two different experimental approaches based on the BLaTM system.
On the methodological side, the economics of the newly developed library-based method may be a useful feature in related future work or in identifying heterotypic TMD-TMD interactions within other biological contexts.
Injuries to the central and peripheral nervous systems remain one of the major health problems worldwide. With the increasing frequency of strokes and traumatic brain injuries, as well as spinal cord injuries in people under 50 years of age and the aging of the population, the problem of finding new strategies for protecting nerve cells is particularly acute.
Unfortunately, there are currently no effective neuroprotectors. The search for drugs that can increase the survival of nerve cells in neurodegenerative conditions is associated with the study of a complex system of regulation of the functioning of genes and proteins.
Intramembrane proteases, such as γ −secretase, typically recruit multiple substrates from an excess of single-span membrane proteins. Other γ-secretase substrates include LDL receptor-related protein, Notch, E-cadherin and ErbB-4. Why did the authors choose APP as a substrate for γ-secretase for research?
For example, DAPT is a γ-secretase inhibitor and indirectly an inhibitor of Notch, a γ-secretase substrate. DAPT may be useful in the study of β-amyloid (Aβ) formation. DAPT has been shown to inhibit Notch signaling in studies of autoimmune and lymphoproliferative diseases, such as ALPS and lupus erythematosus (SLE), as well as in cancer cell growth, angiogenesis, and differentiation of human induced pluripotent stem cells (hIPSC).
DAPT may be considered as a potential drug for stroke treatment. However, DAPT and LY2886721 inhibited the γ-secretase complex containing PS1 rather than the γ-secretase complex with PS2 in humans (in contrast to the results obtained in mice).
It would be interesting, using the authors' approach, to investigate the interaction patterns and potential side effects caused by blocking APP and Notch signaling upon enzyme inhibition.
For example, an important direction will be the development of very selective γ-secretase modulators targeting one subunit of the enzyme.
Authors need to describe in more detail the strengths of their study and the practical implications. Also think about the limitations of this approach and write them out in a separate paragraph Limitations.
The APP protein has been intensively studied since the 1980s due to its central role in the development of Alzheimer’s disease (AD). Its fragment β -amyloid peptide (Aβ) accumulates in amyloid plaques in the brain of AD patients. The processing of APP is involved in the pathogenesis of many neurodegenerative disorders. An increase in the level of APP and amyloid peptide was shown in stroke models, as well as in axonal damage due to injuries of peripheral nerves [https://pubmed.ncbi.nlm.nih.gov/36289917/; https://pubmed.ncbi.nlm.nih.gov/32914392/; https://pubmed.ncbi.nlm.nih.gov/27771897/ https://pubmed.ncbi.nlm.nih.gov/31066008/].
APP is a large transmembrane glycoprotein that crosses the plasma membrane (PM) once. Its large N-terminal domain faces the extracellular environment, while its small C-terminal domain faces the cytoplasm. APP undergoes proteolytic cleavage by α -, β -, and γ-secretases to form several peptides. APP proteolytic products have independent activity and are involved in various cellular processes.
There are amyloidogenic and non-amyloidogenic pathways of APP proteolysis. Amyloidogenic processing of APP occurs in specialized regions of the cell membranes, the lipid rafts. γ-secretase is a large multi-subunit enzyme consisting of presenilin-1 that performs a proteolytic function, presenilin-2 that associates and causes the endoproteolysis of PS1 into the N-terminal fragment and C-terminal fragment (CTF), involved in the substrate recognition of nicastrin, and the anterior pharynx-defective 1 (APH-1) protein that forms a platform for subunit binding . The results of γ-secretase activity include the release of the amyloid peptide Aβ into the extracellular environment promoting the development of AD and the release of the rest into the cytoplasm as the transcription factor AICD is regulating the expression of proapoptotic genes. The components of the γ-secretase complex presenilin 1 and nicastrin involved in APP proteolysis in rat brain neurons and glial cells after photothrombotic stroke [https://pubmed.ncbi.nlm.nih.gov/36289917/].
The results of the work will allow a more selective approach to the search for drugs to restore nerves after injury and reduce disability.
Sharifulina at al. (2022) assumed that axonal APP is concentrated in the caveolar structures of neurons. Caveolin-1 can act independently of caveolae in ischemia. Caveolin can be found in the trans-Golgi network in the cytosol or separate structures, such as caveosomes (early endosomes) and TGN, which can be the site of APP processing toform A β. Thus, the balance of caveolin-1 during ischemia may affect APP processing and the degree of damage to brain cells after ischemia. Caveolin-1 may play an important role in protecting the brain from stroke. Mice with caveolin-1 knockout had less lesions, lower neurological deficits, and less cerebral edema after intracerebral hemorrhage but caveolin-1 knockout mice showed a high level of apoptotic death of penumbra cells after ischemic stroke.
Caveolin-1 level significantly increases in neurons with aging and under oxidative stress. Abrogated Caveolin-1 expression via histone modification regulates brain edema in a mouse model of influenzaassociated encephalopathy [https://pubmed.ncbi.nlm.nih.gov/30670717/]
Caveol-like membrane domains have been characterized in nerve cells. In Alzheimer’s disease, caveolar dysfunction can cause a decrease in α -secretase activity and accumulation of toxic amyloid A peptide. Caveolin-1 is known to physically interact with APP and BACE1, and overexpression of caveolin-1 attenuated γ-secretase-mediated proteolysis of APP and Notch. Caveolin-1 was weakly expressed in rat brain cells, and stroke caused a further decrease in its level [https://pubmed.ncbi.nlm.nih.gov/36289917/] . Caveolae and caveolins may play a key role in APP proteolysis.
In this regard, it would be interesting to study, using the approach of the authors, the interaction of γ-secretases with caveolin-1.
Also, the author should indicate in the Introduction the domain structure of the enzyme and substrate.
Articles that can be supplemented with References:
https://pubmed.ncbi.nlm.nih.gov/35920259/
https://pubmed.ncbi.nlm.nih.gov/32423851/
https://pubmed.ncbi.nlm.nih.gov/36121025/
https://pubmed.ncbi.nlm.nih.gov/32016207/
https://pubmed.ncbi.nlm.nih.gov/34647731/
https://pubmed.ncbi.nlm.nih.gov/37040764/
https://pubmed.ncbi.nlm.nih.gov/36768156/
https://pubmed.ncbi.nlm.nih.gov/36835396/
https://pubmed.ncbi.nlm.nih.gov/33232115/
https://pubmed.ncbi.nlm.nih.gov/30638370/
https://pubmed.ncbi.nlm.nih.gov/30910800/
https://pubmed.ncbi.nlm.nih.gov/33571524/
https://pubmed.ncbi.nlm.nih.gov/30670717/
Author Response
Reviewer 2
We are grateful to the reviewer for carefully reviewing our paper and for bringing up some useful points to improve the manuscript.
No study has yet systematically investigated the interaction between different substrates and an intramembrane protease at the level of their TMDs. Here, authors examined potential pairwise interactions between isolated substrate and γ−secretase TMDs in a biological membrane using two different experimental approaches based on the BLaTM system.
On the methodological side, the economics of the newly developed library-based method may be a useful feature in related future work or in identifying heterotypic TMD-TMD interactions within other biological contexts.
Injuries to the central and peripheral nervous systems remain one of the major health problems worldwide. With the increasing frequency of strokes and traumatic brain injuries, as well as spinal cord injuries in people under 50 years of age and the aging of the population, the problem of finding new strategies for protecting nerve cells is particularly acute.
Unfortunately, there are currently no effective neuroprotectors. The search for drugs that can increase the survival of nerve cells in neurodegenerative conditions is associated with the study of a complex system of regulation of the functioning of genes and proteins.
Intramembrane proteases, such as γ −secretase, typically recruit multiple substrates from an excess of single-span membrane proteins. Other γ-secretase substrates include LDL receptor-related protein, Notch, E-cadherin and ErbB-4. Why did the authors choose APP as a substrate for γ-secretase for research?
Response: In fact, we had started this work by studying interactions of APP due to its relevance for Alzheimer’s disease. As suggested by the reviewer, we had then continued by including other substrates, including the proposed Notch and ErbB4, to test the generality of our findings with APP.
For example, DAPT is a γ-secretase inhibitor and indirectly an inhibitor of Notch, a γ-secretase substrate. DAPT may be useful in the study of β-amyloid (Aβ) formation. DAPT has been shown to inhibit Notch signaling in studies of autoimmune and lymphoproliferative diseases, such as ALPS and lupus erythematosus (SLE), as well as in cancer cell growth, angiogenesis, and differentiation of human induced pluripotent stem cells (hIPSC).
DAPT may be considered as a potential drug for stroke treatment. However, DAPT and LY2886721 inhibited the γ-secretase complex containing PS1 rather than the γ-secretase complex with PS2 in humans (in contrast to the results obtained in mice).
It would be interesting, using the authors' approach, to investigate the interaction patterns and potential side effects caused by blocking APP and Notch signaling upon enzyme inhibition.
Response: We are afraid this would not be feasible with our approach, given that it employs isolated TMDs within reporter protein constructs, rather than functional enzymes.
For example, an important direction will be the development of very selective γ-secretase modulators targeting one subunit of the enzyme.
Authors need to describe in more detail the strengths of their study and the practical implications. Also think about the limitations of this approach and write them out in a separate paragraph Limitations.
Response: This is a good idea. As part of the Discussion and the Conclusions, we have now tried to be more explicit about the strengths and limitations of our approach.
The APP protein has been intensively studied since the 1980s due to its central role in the development of Alzheimer’s disease (AD). Its fragment β -amyloid peptide (Aβ) accumulates in amyloid plaques in the brain of AD patients. The processing of APP is involved in the pathogenesis of many neurodegenerative disorders. An increase in the level of APP and amyloid peptide was shown in stroke models, as well as in axonal damage due to injuries of peripheral nerves [https://pubmed.ncbi.nlm.nih.gov/36289917/; https://pubmed.ncbi.nlm.nih.gov/32914392/; https://pubmed.ncbi.nlm.nih.gov/27771897/ https://pubmed.ncbi.nlm.nih.gov/31066008/].
APP is a large transmembrane glycoprotein that crosses the plasma membrane (PM) once. Its large N-terminal domain faces the extracellular environment, while its small C-terminal domain faces the cytoplasm. APP undergoes proteolytic cleavage by α -, β -, and γ-secretases to form several peptides. APP proteolytic products have independent activity and are involved in various cellular processes.
There are amyloidogenic and non-amyloidogenic pathways of APP proteolysis. Amyloidogenic processing of APP occurs in specialized regions of the cell membranes, the lipid rafts. γ-secretase is a large multi-subunit enzyme consisting of presenilin-1 that performs a proteolytic function, presenilin-2 that associates and causes the endoproteolysis of PS1 into the N-terminal fragment and C-terminal fragment (CTF), involved in the substrate recognition of nicastrin, and the anterior pharynx-defective 1 (APH-1) protein that forms a platform for subunit binding . The results of γ-secretase activity include the release of the amyloid peptide Aβ into the extracellular environment promoting the development of AD and the release of the rest into the cytoplasm as the transcription factor AICD is regulating the expression of proapoptotic genes. The components of the γ-secretase complex presenilin 1 and nicastrin involved in APP proteolysis in rat brain neurons and glial cells after photothrombotic stroke [https://pubmed.ncbi.nlm.nih.gov/36289917/].
The results of the work will allow a more selective approach to the search for drugs to restore nerves after injury and reduce disability.
Sharifulina at al. (2022) assumed that axonal APP is concentrated in the caveolar structures of neurons. Caveolin-1 can act independently of caveolae in ischemia. Caveolin can be found in the trans-Golgi network in the cytosol or separate structures, such as caveosomes (early endosomes) and TGN, which can be the site of APP processing toform A β. Thus, the balance of caveolin-1 during ischemia may affect APP processing and the degree of damage to brain cells after ischemia. Caveolin-1 may play an important role in protecting the brain from stroke. Mice with caveolin-1 knockout had less lesions, lower neurological deficits, and less cerebral edema after intracerebral hemorrhage but caveolin-1 knockout mice showed a high level of apoptotic death of penumbra cells after ischemic stroke.
Caveolin-1 level significantly increases in neurons with aging and under oxidative stress. Abrogated Caveolin-1 expression via histone modification regulates brain edema in a mouse model of influenzaassociated encephalopathy [https://pubmed.ncbi.nlm.nih.gov/30670717/]
Caveol-like membrane domains have been characterized in nerve cells. In Alzheimer’s disease, caveolar dysfunction can cause a decrease in α -secretase activity and accumulation of toxic amyloid A peptide. Caveolin-1 is known to physically interact with APP and BACE1, and overexpression of caveolin-1 attenuated γ-secretase-mediated proteolysis of APP and Notch. Caveolin-1 was weakly expressed in rat brain cells, and stroke caused a further decrease in its level [https://pubmed.ncbi.nlm.nih.gov/36289917/] . Caveolae and caveolins may play a key role in APP proteolysis.
In this regard, it would be interesting to study, using the approach of the authors, the interaction of γ-secretases with caveolin-1.
Response: Well, caveolin-1 does not contain a TMD whose potential interaction with g-secretase TMDs could be tested. Testing potential interactions between soluble domains is beyond the scope of this work.
Also, the author should indicate in the Introduction the domain structure of the enzyme and substrate.
Response: Sure, we have now described the transmembrane topology of presenilin and substrates at the beginning of Results to better illuminate the design of the project.
Articles that can be supplemented with References:
https://pubmed.ncbi.nlm.nih.gov/35920259/
https://pubmed.ncbi.nlm.nih.gov/32423851/
https://pubmed.ncbi.nlm.nih.gov/36121025/
https://pubmed.ncbi.nlm.nih.gov/32016207/
https://pubmed.ncbi.nlm.nih.gov/34647731/
https://pubmed.ncbi.nlm.nih.gov/37040764/
https://pubmed.ncbi.nlm.nih.gov/36768156/
https://pubmed.ncbi.nlm.nih.gov/36835396/
https://pubmed.ncbi.nlm.nih.gov/33232115/
https://pubmed.ncbi.nlm.nih.gov/30638370/
https://pubmed.ncbi.nlm.nih.gov/30910800/
https://pubmed.ncbi.nlm.nih.gov/33571524/
https://pubmed.ncbi.nlm.nih.gov/30670717/
Response: Sure, these are good papers. However, the research literature on g-secretase and its substrates is extremely rich and diverse and we have tried to focus on those citations that cover the structural aspects related to the present work.
Reviewer 3
This manuscript presents the interactions between TMDs and g-secretases. The writing is straightforward and it attracted to the readership of the International Journal of Molecular Sciences; however, minor revision of the manuscript is necessary prior to publication.
We are grateful to the reviewer for carefully reviewing our paper and for bringing up some useful points to improve the manuscript.
Comments:
- Please revise last paragraph of section 2.1.
Response: done!
- The resolution of all figures should be improved.
Response: a limited resolution arises from copying TIFF files into Word. However, the original TIFF files – to be uploaded soon – are of high resolution.
- In this manuscript, the authors only focuses on the interaction between TMDs, including APP, and g-secretases. But, it would be better to present how the enzymatic activity of g-secretases is altered upon interaction with other TMDs.
Response: Testing the impact of the TMDs on enzymatic activity is a project that is very different from the goals associated with our current work, which focuses on substrate recognition. This proposition is thus regarded beyond the scope of the current work.
Comments on the Quality of English Language
The English should be edited in some points. For example, the last paragraph of section 2.1
Response: done!
Reviewer 3 Report
This manuscript presents the interactions between TMDs and g-secretases. The writing is straightforward and it attracted to the readership of the International Journal of Molecular Sciences; however, minor revision of the manuscript is necessary prior to publication.
Comments:
1. Please revise last paragraph of section 2.1.
2. The resolution of all figures should be improved.
3. In this manuscript, the authors only focuses on the interaction between TMDs, including APP, and g-secretases. But, it would be better to present how the enzymatic activity of g-secretases is altered upon interaction with other TMDs.
The English should be edited in some points. For example, the last paragraph of section 2.1.
Author Response
Reviewer 3
This manuscript presents the interactions between TMDs and g-secretases. The writing is straightforward and it attracted to the readership of the International Journal of Molecular Sciences; however, minor revision of the manuscript is necessary prior to publication.
We are grateful to the reviewer for carefully reviewing our paper and for bringing up some useful points to improve the manuscript.
Comments:
- Please revise last paragraph of section 2.1.
Response: done!
- The resolution of all figures should be improved.
Response: a limited resolution arises from copying TIFF files into Word. However, the original TIFF files – to be uploaded soon – are of high resolution.
- In this manuscript, the authors only focuses on the interaction between TMDs, including APP, and g-secretases. But, it would be better to present how the enzymatic activity of g-secretases is altered upon interaction with other TMDs.
Response: Testing the impact of the TMDs on enzymatic activity is a project that is very different from the goals associated with our current work, which focuses on substrate recognition. This proposition is thus regarded beyond the scope of the current work.
Comments on the Quality of English Language
The English should be edited in some points. For example, the last paragraph of section 2.1
Response: done!